# The Role of the Thalamus in Post-Traumatic Stress Disorder

**DOI:** 10.3390/ijms22041730

**Published:** 2021-02-09

**Authors:** Takanobu Yoshii

**Affiliations:** 1Department of Psychiatry, Graduate School of Medical Science, Kyoto Prefectural University of Medicine, Kamigyo-ku, Kyoto 602-8566, Japan; takanon@koto.kpu-m.ac.jp; 2Kyoto Prefectural Rehabilitation Hospital for Mentally and Physically Disabled, Naka Ashihara, Johyo City, Kyoto 610-0113, Japan

**Keywords:** PTSD, thalamus, fMRI, morphology, EMDR

## Abstract

Post-traumatic stress disorder (PTSD) has a high lifetime prevalence and is one of the more serious challenges in mental health care. Fear-conditioned learning involving the amygdala has been thought to be one of the main causative factors; however, recent studies have reported abnormalities in the thalamus of PTSD patients, which may explain the mechanism of interventions such as eye movement desensitization and reprocessing (EMDR). Therefore, I conducted a miniature literature review on the potential contribution of the thalamus to the pathogenesis of PTSD and the validation of therapeutic approaches. As a result, we noticed the importance of the retinotectal pathway (superior colliculus−pulvinar−amygdala connection) and discussed therapeutic indicators.

## 1. Introduction

Post-traumatic stress disorder is a common mental disorder, with high lifetime prevalence of approximately 6–10% [1,2]. The prevalence of PTSD in trauma-exposed people has been approximately 20% [3]. PTSD is induced by traumatic stress including life threatening, actual or threatened severe injury, and sexual violence. In DSM-V criteria [4], PTSD has the following symptoms: the intrusion of unwanted memory updates related to traumatic stress, avoidance for reminders, negative alterations in mood, and hyper-arousal. Conservatively, fear-conditioned learning involving the amygdala has been considered one of the causative factors. Prolonged exposure therapy is an established approach designed to reduce PTSD symptoms and related problems (e.g., depression, anger, guilt) via in vivo imaginal exposure to traumatic memory [5]. The therapeutic mechanism of PE is considered the alteration of functional connectivity between the amygdala, hippocampus, and frontal cortical regions [6]. However, it has not resulted in therapeutic breakthrough because of its unpleasantness, resulting in a dropout rate of at least 50% [7]. Eye movement desensitization and reprocessing (EMDR) has also been developed as a therapeutic approach, but the mechanism of desensitization using eye movement is as yet unclear. There is debate about the precise mechanism by which EMDR appears to relieve PTSD symptoms may simply be a variety of exposure therapy [8].

Volumetric neuroimaging studies of PTSD have identified different atrophic areas, such as the hippocampus, anterior cingulate cortex (ACC), posterior cingulate cortex [9], insular cortex [10], orbitofrontal cortex [11], ventromedial prefrontal cortex [12], occipital cortex [13], calcarine sulcus [14], or amygdala [15]. It is difficult to determine which region is most important because it is unclear whether the traumatic stress or the predisposition of the patient contributes more strongly to the pathology. Although brain volume reduction in the thalamus has rarely been reported in research targeting PTSD, brain atrophy in the bilateral thalamus has been reported in pain study [16]. It was reported that psychological torture induced pain [17], and pain itself should be considered as one of the stress contents. It is difficult to rule out that thalamus has the possibility of volumetric change induced by stress, and our group, in fact, observed stress-induced brain atrophy in the thalamus [18].

The thalamus was originally thought to act as a hub for relaying sensory information to the cortices and occasionally playing a role in processing this information. However, this view has gradually changed, and the thalamus is now considered to have many functions, acting as a sensory hub between other subcortical nuclei and the cortices and contributing to sleep and wake awareness, motor control, and cognition. Thalamic functional abnormalities are thought to contribute to the dysregulation of sensory filtering, circadian rhythms, levels of alertness, and consciousness [19]. It has been mentioned that fearful stimulation activates the thalamus [20]. In the context of fear-related learning, sensory processing, including visual processing in the thalamus, has been investigated with respect to its impact on amygdala function and output, rather than as an important psychological or pathophysiological process that shapes the development of PTSD [21]. However, our animal voxel-based morphometry (VBM) study revealed the induction of brain atrophy in the thalamus by severe stress [18]. In addition, recent research work revealed the efficacy of interventions via visual technique: visual neurofeedback using implicit fear exposure [22,23] and visual game task [24]. Therefore, I consider the thalamus to be a promising area for stress research and thus conducted a mini literature review of the contribution of the thalamus to the pathogenesis of PTSD.

## 2. Results

This paper was initially designed as a systematic review. According to a result of literature research, the importance of the retinotectal pathway was recognized. At first, according to my bibliographic literature research, I found 132 studies mentioning PTSD and the thalamus. This total was reduced when I used narrower search terms: the number of studies mentioning “PTSD AND thalamus AND MRI” was 63. Unfortunately, I was not able to find sufficient numbers of volumetric studies (two studies with “PTSD AND thalamus AND MRI AND atrophy”, three with “PTSD AND thalamus AND DTI”) (Figure 1) and I also researched nine with “Childhood maltreatment AND MRI” squeezed to four papers volumetric study (Figure 2). I also read through 33 studies mentioning “PTSD AND thalamus AND MRI AND fMRI” and three studies with “PTSD AND thalamus AND MRI AND DTI” (Figure 1). Throughout the course of this study, I noticed the importance of retinotectal pathways and researched 12 including “PTSD AND colliculus” and two including “PTSD AND pulvinar”. I read through nine articles of them (Figure 3). I also aimed this research to bridge therapeutic interventions and researched 14 studies including “PTSD AND thalamus AND therapy” and I read through four reports of them (Figure 4). Tables 1–4 summarize the articles I included and provide a brief interpretation of each. The results of the review, separated according to subject area, are described below.

## 3. Stress-Related Structural Change in the Thalamus

### 3.1. Volume Change in Thalamus

There are few reports that discuss stress-induced brain atrophy in the thalamus, and there is a report mentioning the trait of volume reduction in the thalamus which failed to reach significance [26,27]. Meta-analysis study of VBM via large-scale neuroimaging consortium study on PTSD did not reach significance in the thalamus volume [28]. On the other hand, volume loss in the right thalamus was detected in PTSD malingerers [29].

However, there is a negative correlation between thalamic volume and re-experiencing in PTSD [30]. Although numbers of literature targeting PTSD have been small, a meta-analysis study mentioned brain atrophy in the bilateral thalamus [16]. It is difficult to separate mental stress from physical pain because chronic pain is sometimes contingent on the psychological toll of torture [17]. I consider that pain is probably one of the most important sources of physical and even psychological stress. The volume of the thalamus in the researches targeting childhood maltreatment has been studied; however, the volumetric changes in childhood maltreatment may be inconsistent. There is a negative correlation between childhood maltreatment score and the volume of the right thalamus [31], and there is a positive correlation with the volume of the left thalamus [32]. There is also a study on the volume reduction of the bilateral thalamus in physical childhood maltreatment [33], which may be a match to the result of pain studies, and it has been discussed stress paradigms may influence the volume variation in the thalamus [32]. However, no associations between age of exposure and volume of the thalamus are mentioned in other studies [34]. In this bibliographic literature research, meta-analysis study indicating volume reduction of thalamus both in PTSD and childhood maltreatment was not detected. I elucidate the possibility that the variation of volume-change from stress paradigm may dampen the statistical power for meta-analysis.

In an animal model, chronic stress induced dendritic atrophy in the medial thalamic nuclei, whose relationship to fear conditioning has been investigated [35]. It has been reported that a thinner right prefrontal cortex and larger right thalamus have been found to be related to denial and response prevention [36]. An animal VBM study of the response to a single prolonged stress factor indicated significant atrophy in the ventral lateral nuclei in the thalamus [18]. Generally, atrophic effect size of stress must be smaller than that of endogenous disease. Therefore, I consider that it is not sufficient to exclude participants with endogenous disease as a confounding factor for brain atrophy in clinical research. The summary of this literature study is described in Table 1.

### 3.2. Diffusion Tensor Imaging (DTI)

There are only a small number of studies that mention the abnormality of the thalamus in PTSD patients, and further research is required to determine whether the abnormality is in the white matter or the anatomical connectivity within the thalamus, or both. A DTI study reported a reduction in mean diffusivity (MD) and axial diffusivity (AD) in the right thalamus in PTSD patients [37]. The bilateral dorsal cingulum and right anterior corona radiation display low fractional anisotropy (FA) in PTSD patients, and low FA in the anterior corona radiation is correlated with PTSD severity, suggesting the relevance of the integrity of the limbic-thalamo-cortical tracts to PTSD [38]. It was reported that FA value improvement was detected in the anterior corona radiation and right thalamus of recovered fibromyalgia with PTSD symptom [39]. I was not able to detect any meta-analysis study detecting abnormality white matter abnormality around thalamus in PTSD; however, there is a meta-analysis study which reports the reduction of FA in the left anterior thalamic radiation and bilateral fornix in survivors in childhood maltreatment [40]. A recent research study mentions that thalamic volume is related to increased anterior thalamic radiations in children with childhood maltreatment [41]. The discussion is similar to the previous paragraph, but I reason that the meta-analysis may not reach the level of statistical significance because of the influence of stress content. See Table 1 as the summary of literature research.

### 3.3. Depleted Regional Activity in the Right Thalamus

It has been mentioned that PTSD patients display depleted functional activity in the thalamus [42,43,44], and the right side of the thalamus is thought to have pathological significance [45], with psychotherapy increasing activity in the thalamus during memory retrieval [46]. PTSD patients showed enhanced activity of the thalamus by seeing neutral (not traumatic) pictures [47]. In addition, resting-state functional MRI scans of PTSD patients have revealed enhanced regional connectivity in the thalamus [45,48,49]. Such patients exhibited depleted cluster coefficients within the bilateral thalamus, representing a fraction of all possible connections that connect the neighbors of a given node, in an analysis of gray matter structural connectivity [50]. PTSD patients showed strong positive correlations between the blood oxygen level–dependent (BOLD) signal and symptom severity in both the thalamus and the head of the caudate nucleus [51]. It has also been reported that the severity of early life stress is positively correlated to global-based connectivity in the thalamus [52], and childhood maltreatment is thought to affect stress resilience. Resilience scores were positively correlated with BOLD signal strength in the right thalamus [53]. In addition, PTSD with dissociative symptoms (severe PTSD) is associated with higher activation of the left thalamus for nonconscious fear than nondissociative PTSD [54]. The summary of this literature research described in Table 1.

In conclusion, I consider that there is laterality in the pathological significance of the thalamus for PTSD, and this may explain the discrepancies between reports with respect to thalamic activity. Although the mechanism of pathological laterality in PTSD is not clear, it has been known that visual perception and spatial awareness are right-side-dominant, and this laterality may therefore occur because the pathogenesis of PTSD is dependent on visual experience. With the progression of the disease, however, abnormalities may also appear in the left thalamus.

### 3.4. Connectivity Research

Functional connections between the thalamus and amygdala, and between the thalamus and the anterior cingulate cortex (ACC), have been emphasized in PTSD researches. Increased co-activity ACC, posterior cingulate cortex, and thalamus is detected in PTSD patients [55]. PTSD can cause widespread increases in the activation of effective connectivity between the thalamus and the amygdala, striatum, rostral ACC, and ventral occipital cortex, all of which encode scenes and could cause flashbacks [56]. Both resting-state [57] and in-task [58,59] functional connectivity between the amygdala and the thalamus have been reported, and in-task amygdala–thalamus connectivity is correlated with PTSD severity [58]. Some studies have shown that functional connectivity between the ACC and the thalamus is depleted in PTSD patients [19,52,60,61], and emotional processing is particularly strongly associated with the coactivity of the ACC and the posterior cingulate cortex (PCC), as mediated by the thalamus [55].

The severity of PTSD may increase functional connectivity between the thalamus and other sensory areas. Dissociative symptoms have been considered to indicate severe PTSD, and dissociated PTSD patients show depleted connectivity between the left ventrolateral thalamus (VLT) and the left superior frontal gyrus, right parahippocampal gyrus, and right superior occipital gyrus. In contrast, they exhibit enhanced connectivity between the VLT and the right insula, right middle frontal gyrus, superior temporal gyrus, right cuneus, and left parietal lobe [62]. The connectivity between the pedunculopontine nuclei, as part of the reticular activation system, and the anterior nucleus of the right thalamus is negatively correlated with dissociative symptoms (derealization and depersonalization) [48].

The progression of PTSD may have brought enhancements in connections between thalamus and autonomic systems. Enhanced functional connectivity between the thalamus and the locus coeruleus has also been reported in PTSD [63]. Simultaneous enhancement of neural activity in the periaqueductal nuclei and in the midline thalamic nuclei has been reported in an animal PTSD model [64]. The thalamus is involved in the central autonomic network (CAN), and low in-task heart rate variability (HRV) has been proposed as a biomarker of PTSD; however, the covariation between brain connectivity related to the CAN and HRV is diminished in PTSD patients [65]. High responders to stress exhibit local brain atrophy in the ventral tegmental area (VTA) and enhanced connectivity between the VTA and the thalamus [66]. The alexithymia scores (Toronto Alexithymia Scale 20) of PTSD patients were positively correlated with suicide ideation [67] and there is the positive correlation between the alexithymia scores and the response in the thalamus in PTSD patients [68]. The summary of this literature research is described in Table 2.

**Table 1 ijms-22-01730-t001:** The summary of literature study for structure and regional activity in the thalamus.

Volume Change in Thalamus	Reference Number
Volume reduction (not significant)	[26,27,28]
Volume loss in the right thalamus was detected in PTSD malingerers	[29]
Negative correlation between volume of right thalamus and re-experiencing	[30]
Volume loss in the bilateral thalamus in pain meta-analysis	[16]
Negative correlation between right thalamic volume and childhood maltreatment	[31]
Positive correlation between left thalamic volume and childhood maltreatment	[32]
Volume reduction in bilateral thalamus in cases of childhood physical maltreatment	[33]
No associations between trauma exposure age and volume of thalamus	[34]
Volume loss in the bilateral thalamus (ventrolateral nuclei) in animals under severe stress	[18]
Thinner right prefrontal cortex and larger right thalamus are related to denial and response prevention in PTSD	[36]
**Diffusion Tensor Imaging (DTI)**	
Loss of MD and AD in right thalamus	[37]
Low FA in bilateral dorsal cingulum and anterior corona radiate	[38]
FA value improvement in anterior corona radiation and right thalamus in recovered PTSD patients	[39]
Increased anterior thalamic radiation via childhood maltreatment correlated to thalamic volume	[40]
**Depleted Regional Activity in the Right Thalamus**	
Depleted regional activity	[42,43,44,45]
Enhanced regional connectivity within thalamus	[45,48,49]
Psychotherapy increased activity in thalamus during retrieval	[46]
Enhanced activity of thalamus from showing PTSD patients a neutral picture	[47]
Depleted cluster coefficients within bilateral thalamus	[50]
BOLD signal positively correlated with symptoms	[51]
Early life stress severity positively correlated with connectivity in thalamus	[52]
Resilience score is positively correlated with BOLD signal in right thalamus in cases of childhood maltreatment	[53]
Laterality of activation (pathological significance of right side)	[46]
Enhanced activity in left thalamus during dissociation	[54]

### 3.5. The Retinotectal Pathway in Fear-Related Learning in the Thalamus

I believe that the retinotectal pathway plays an important role in the progression of PTSD pathogenesis. Visual information bypassed from geniculostriate system via the superior colliculus (SC) projecting to the pulvinar as the retinotectal pathway. There are some reports that SC itself directly plays a role combined with the periaqueductal gray as an innate alarm system modulating defensive behavior [69] and activating in the social eye-contact situation in PTSD patients [70,71]. Subliminal threat induces neural activity in SC and periaqueductal gray in PTSD patients [72]. PTSD with dissociative subtype, compared to PTSD without dissociation, increased resting state connectivity between SC and the right dorsal lateral prefrontal cortex [73]. The pulvinar is a posterior part of the thalamus and constitutes the retinotectal pathway, which is separate from the geniculostriate system. A recent study demonstrated that the connectivity between pulvinar and V1 contributed to fear anticipation [74]. Patients with pulvinar lesions exhibit disrupted implicit fear-related visual processing [75]. The geniculostriate system contributes to form (ventral stream) and space (dorsal stream) perception in visual processing. Both the geniculostriate and the retinotectal pathways contribute to visual processing, and the latter in particular mediates implicit processing of fearful stimuli [76,77]. Further research is needed to clarify the function of the retinotectal pathway; however, it has been demonstrated that this circuit contributes to fear learning [78].

This circuit may be necessary for the development of PTSD. It has been reported that trauma survivors with a smaller pulvinar exhibit lower morbidity rates for PTSD [79], and fMRI studies indicated enhanced connectivity between the SC and the ACC in PTSD patients [63]. In addition, depletion of left pulvinar seed functional connectivity to sensory regions (left superior parietal lobule, left middle temporal gyrus, and right postcentral gyrus) in PTSD patients [80] and depletion of right pulvinar seed connectivity to primary visual and higher sensory regions (left superior frontal gyrus, left superior parietal lobule, bilateral precuneus, right inferior parietal lobule, right precentral gyrus medial segment) in dissociative PTSD patients was reported [80]. Therefore, I consider that the retinotectal pathway is likely to play a key role in fear conditioning, and disruption for pulvinar function may have a preventative effect toward PTSD morbidity. The summary of this literature research is presented in Table 3.

**Table 2 ijms-22-01730-t002:** The summary of literature study for connectivity in the thalamus.

Connectivity Research	Reference Number
Increased coactivity with ACC, posterior cingulate cortex, and thalamus	[55]
Increase in effective connectivity from thalamus to amygdala	[56,57,58,59]
Increase in effective connectivity from thalamus to ACC, striatum, and occipital cortex	[56]
Positive correlation between thalamus−amygdala and PTSD severity	[58]
Depletion of connectivity between thalamus and ACC	[19,52,60,61]
Emotional processing correlation between thalamus and ACC/PCC	[55]
Alteration of connectivity from VLT to other sensory areas in dissociative PTSD patients	[62]
Enhanced connectivity of pedunculopontine nuclei (reticular activation system) and anterior thalamic nucleus in dissociative PTSD	[48]
Enhanced connectivity between thalamus and locus coeruleus	[63]
Simultaneous enhancement of activity in midline thalamus and periaqueductal nuclei in animal PTSD model	[64]
Diminished correlation between CAN and HRV	[65]
Enhanced connectivity between thalamus and VTA	[66]
Positive correlation between thalamus activity and alexithymia	[68]

## 4. Discussion on Therapeutic Implications Targeting the Thalamus

Based on this literature review, I believe that the retinotectal–pulvinar pathway and thalamosensory connectivity contribute to the development of PTSD, and that depletion in these regions before stress exposure may have preventative effects against worsening PTSD. In fact, a recent clinical study reported that trauma-exposed control patients without worsening PTSD exhibited depleted resting-state functional connectivity between the thalamus and the postcentral gyrus, while both healthy controls and PTSD patients did not exhibit such depletion [81]. A smaller pulvinar in similar traumatized control patients has been also reported [79]. Defective pulvinar functioning may thus result in a failure of fear processing [75], and I believe that reduced pulvinar function may contribute to prevent the development of PTSD. On the other hand, the functional amygdala seed connectivity study in PTSD indicates the enhancement of connectivity between centromedial amygdala and pulvinar, although the depletion between basolateral amygdala and SC was detected [82]. In addition, recovered PTSD patients showed that the lower tract strength of the amygdala–thalamus connection was normalized during recovery, while that of amygdala–hippocampus connection remained low [83]. PTSD patients also display disrupted regional activity of the thalamus, including the dorsal medial thalamus, and future research should investigate whether this may hinder overwriting traumatic memories with memories of ordinary daily life. The summary of literature research for therapeutic implications is described in Table 4.

Taken together, I assume promising research: first, suppressing this pathway as a direction of preventing the progression of PTSD; second, activating retinotectal pathway followed with minimal stress exposure paradigm for desensitization of thalamus amygdala connectivity; and third, activating geniculovisual cortex pathways to reduce contribution of retinotectal pathway in visual fear processing.

In addition, I also would like to bridge the physiology of the thalamus to therapeutic implications and consider the contribution of the physiology of thalamus to proposed therapeutic interventions.

### 4.1. EMDR

EMDR can be described as a methodology that extends PE. It is thought to take advantage of the fact that eye movements facilitate learning, but the mechanism has long been unclear, and many EMDR researchers themselves have relied on cognitive behavioral therapy as the basis for its effectiveness. However, a recent animal study revealed that projection from the SC to the mediodorsal thalamus may contribute to the processing of conditioned fear [78], and especially mediodorsal thalamus may contribute in reactions to fear memory [64,84,85,86]. Saccade eye movement is used for the desensitization of traumatic memory in EMDR therapy, and a recent animal study demonstrated that visual bilaterally alternating sensory stimulation, such as EMDR stimulation, provided a fear-reducing effect, with sustained activation from the SC to the mediodorsal thalamus [87]. This result may partly explain the therapeutic mechanism of EMDR. Despite the finding that limited pulvinar function may exert a preventative effect toward PTSD, the EMDR animal model exhibited enhanced functioning of the SC and mediodorsal thalamus. The SC and mediodorsal thalamus activation may have a therapeutic effect by promoting the exposure and desensitization of the thalamus–amygdala complex. Following EMDR treatment, patients also showed a significant reduction in gray matter volume in the left thalamus region [88], suggesting that the treatment may have modified the laterality that was induced by PTSD. I consider that the efficacy of EMDR for treating PTSD supports the idea that the thalamus may be the key site of PTSD pathogenesis. See Figure 5 as a schematic summary.

### 4.2. Functional MRI Neurofeedback Technique

A therapeutic approach targeting unconsciousness has been proposed [7], and one that attempts to reprogram the unconscious using a fMRI-based technique called decoded neurofeedback (DecNef) has been tried [22,23]. In conclusion, I could not find any literature which indicates the contribution of thalamic change in this method. Tasks involving the unconscious are complicated: they require computational calculation of brain activity in the higher visual cortex during fear and control stimuli that attempt to stimulate brain activity in the higher visual cortex to approximate that observed during fear stimuli without using direct visual stimulation as non-explicit fear stimulation. This approach may have been inspired by prolonged exposure therapy, but the activity of the amygdala does not appear to be significantly altered in the learning phase. Only an acute BOLD signal response in the amygdala under a fear-target confrontation was demonstrated [22]. Based on our belief that the retinotectal pathway is more important than the geniculovisual cortex pathway in the processing of visual fears, I assume that this intervention achieves therapeutic benefits by reducing the contribution of the retinotectal pathway.

### 4.3. Visual Task Games

The visiospatial game task followed with minimal exposure has been applied for prevention of PTSD via proof randomized control trial [24]. There is a debate that the therapeutic mechanisms via competition for limited working memory resources for the task [90] and I also could not find any literature describing the contribution of thalamic change for this intervention. Although these studies were not designed for action on the thalamus, it seems difficult to rule out that it is undergoing some action on the thalamus via visual task. At present, there is no evidence that the thalamus is affected by this task; however, it is certainly a task including discrimination and spatial cognition, and it can be reasonably speculated that the geniculovisual pathway would be used. It is interesting whether this intervention might produce preventative effect via reducing the contribution of retinotectal pathway in the phase of PTSD progression.

### 4.4. Hyperbaric Oxygen Therapy (HBOT)

I would like to introduce an intervention method that is neither medications nor exposure. It has been shown in a randomized control trial that treating patients with a history of childhood sexual abuse with HBOT results in an improvement of PTSD symptoms and psychological distress, as well as an improved FA value (based on MRI-DTI) in the anterior thalamic radiation, left thalamus, and left insula [39]. This treatment was originally designed to treat fibromyalgia; however, it was subsequently suspected to have activated the entire thalamus and may thus have additional therapeutic effects for PTSD. HBOT may also promote the exposure and desensitization of thalamus–amygdala complex. This treatment is expected to have few serious side effects, and I consider that HBOT is thus a promising intervention for PTSD, which requires further research.

### 4.5. Oxytocin Administration

Oxytocin is a neurochemical agent that is thought to improve human interaction and has been experimentally applied to PTSD therapy. Oxytocin administration enhanced the activity of the left thalamus in both patients and controls during a distraction task (resetting negative feelings with a working-memory task) and was negatively correlated with error rates in this task in PTSD patients [89]. Oxytocin administration also decreased the functional connectivity between the left thalamus and the amygdala in male PTSD patients and trauma-exposed controls, although female patients indicated enhanced connectivity between the left thalamus and the amygdala [89]. The mechanism of influencing on thalamus amygdala connectivity with sexual variation has been unknown and the potential efficacy of oxytocin administration should be further investigated.

### 4.6. Other Medications

Conventionally used medications may have some effect on the thalamus, but these interventions do not yield satisfactory therapeutic effects. Several studies have thus indicated a relationship between the thalamus and PTSD. These findings provide only a seed for future research and are not sufficient to establish any new interventions. Selective serotonin reuptake inhibitors (SSRIs) have been established as a treatment option for PTSD, and the mediodorsal thalamus is a serotonin-rich area. Although one animal study showed that chronic predator-scent exposure altered serotonin and dopamine levels in the thalamus [91], another did not show a noticeable effect in the thalamus [92]. Prazosin: α -1 adrenergic antagonist treatment has been already established as intervention to hyper arousal and nightmares in PTSD [93]. Oral propranolol administration (1 mg/kg) improved PTSD symptoms following the enhancement of activation of the thalamus and amygdala [94]. Noradrenergic system is intimately connected to the thalamus, and it is an interesting and promising area for future research seeking the relationship between the noradrenergic modulation and alteration of the thalamus in PTSD patients. A significant correlation between re-experiencing and thalamic β_2_ nicotinic acetylcholine receptor binding using single-photon emission computed tomography (SPECT) has already been described [95], The micro-opioid system in the dorsomedial thalamus has also been found to contribute to fear extinction [96]. However, these have not yet been targeted in medication research.

## 5. Methods

I conducted a bibliographic search in PubMed using the search terms “PTSD AND thalamus AND MRI AND atrophy”, “PTSD AND thalamus AND MRI AND DTI”, “PTSD AND thalamus AND MRI AND fMRI”. To include research other than MRI studies. There are a few numbers for structural studies of abnormality in the thalamus and I added “childhood maltreatment and MRI and atrophy” and “childhood maltreatment and MRI and volume”. In addition, because the purpose of this review was to investigate established treatment methods (especially EMDR) and explore new treatment approaches, I used the search terms “PTSD AND thalamus AND therapy”. Finally, I applied the term “Not injury” to the selection, and I excluded conference papers and studies not published in English (Figure 1). After this research, we noticed the importance of the retinotectal pathway on PTSD additional bibliographic search was carried out, “PTSD AND colliculus” and “PTSD AND pulvinar”. These literature searches conform to the PRISMA guidelines [25].

### Limitations

This article was initially planned as a systematic review for the relationships between thalamus and PTSD and the contribution of retinotectal pathways for fear processing was noticed. I aimed to bridge these findings to therapeutic mechanism; however, a few articles were detected for the therapeutic contribution of thalamus in PTSD, especially for neurofeedback techniques and visual task game. Although this may indicate a vanguard of the therapeutic research targeting thalamus, I had to seek additional literature depending upon the author’s experience in order to deepen the section of discussion. Therefore, this may violate the validity and reproducibility of the discussion in this article.

## 6. Conclusions

Structural and volumetric researches detecting abnormalities in the thalamus have not been sufficiently accumulated, and there may be problems categorizing stress contents which might have caused failure to detect structural abnormalities in thalamus. However, animal studies demonstrated that the SC–pulvinar–mediodorsal thalamus–amygdala pathways, contribute to visual fear processing. Based on this literature review, I propose that the activation of this system promotes the exposure and desensitization of the thalamus–amygdala complex. It seems paradoxical that PTSD patients often require exposure techniques for recovery, although many patients with acute stress disorder recover spontaneously. A functional defect in an atrophy of the thalamus may be related to the failure of desensitization of the thalamus–amygdala complex by exposure experienced in everyday life. In addition, with the spread of COVID-19, the treatment of PTSD needs to address the ongoing disaster. We believe that it would be promising to study interventions for this site as preventive approaches for COVID-19 induce PTSD.

## Figures and Tables

**Figure 1 ijms-22-01730-f001:**
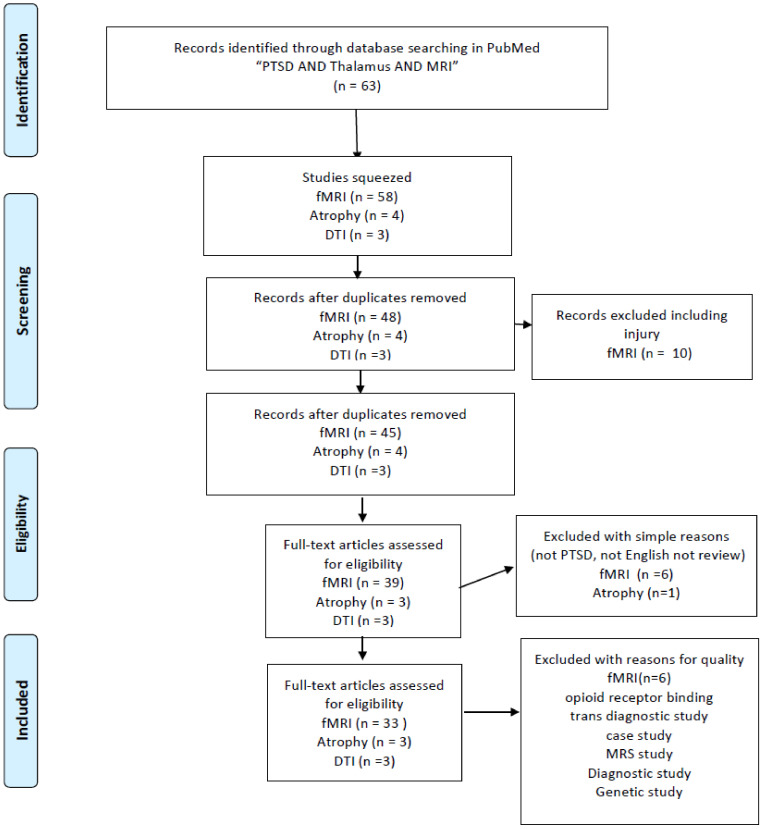
PRISMA flow diagrams [25] of literature research for PTSD neuroimaging research targeting on the thalamus.

**Figure 2 ijms-22-01730-f002:**
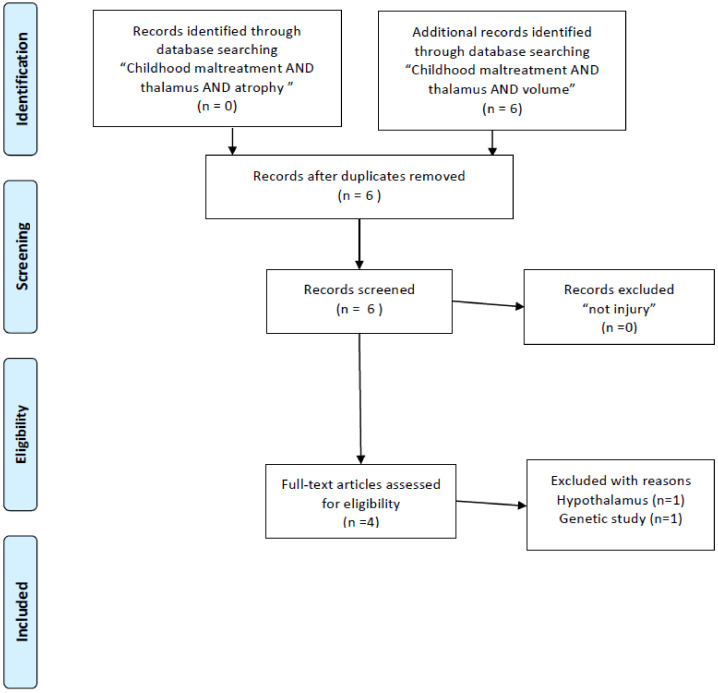
PRISMA flow diagrams [25] of literature research for brain structure research of childhood maltreatment targeting on the thalamus.

**Figure 3 ijms-22-01730-f003:**
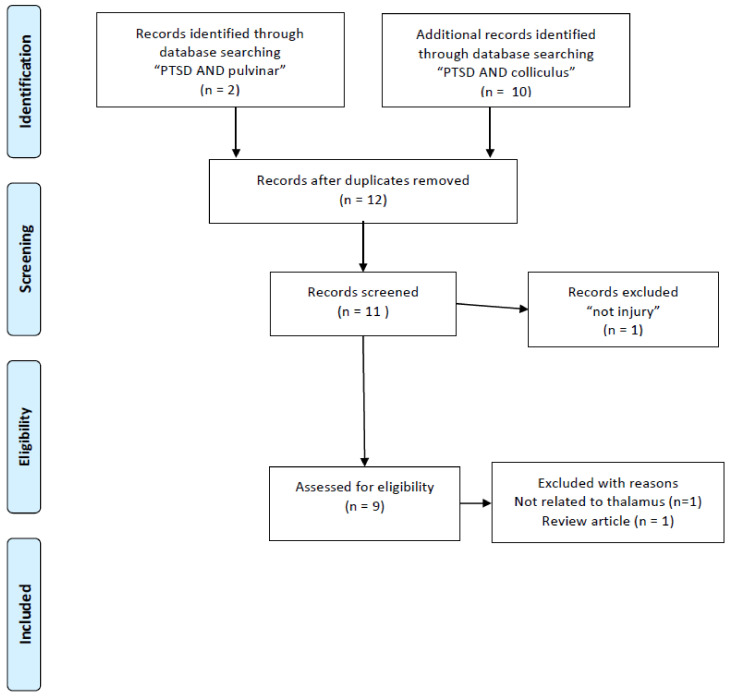
PRISMA flow diagrams [25] for literature research for PTSD targeting retinotectal pathways.

**Figure 4 ijms-22-01730-f004:**
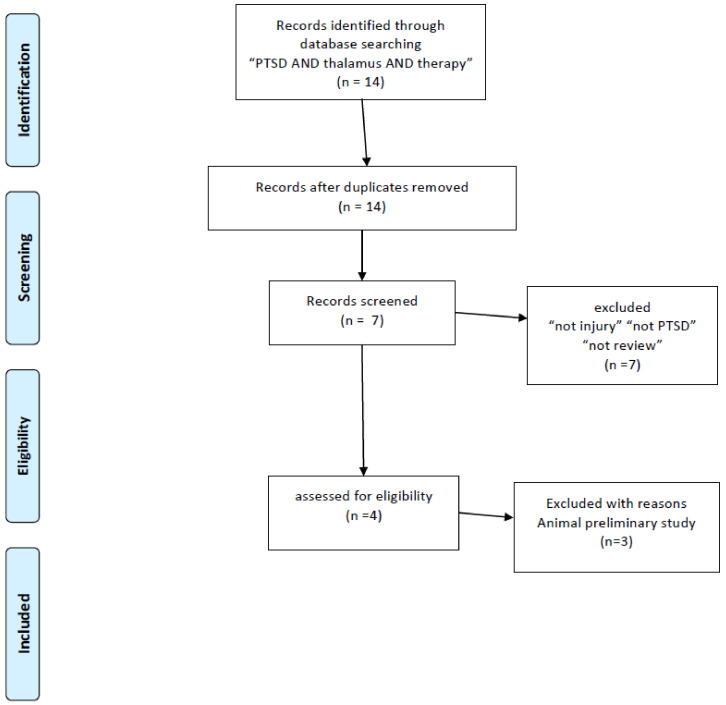
PRISMA flow diagrams [25] of literature research for contribution of thalamus physiology to therapy in PTSD.

**Figure 5 ijms-22-01730-f005:**
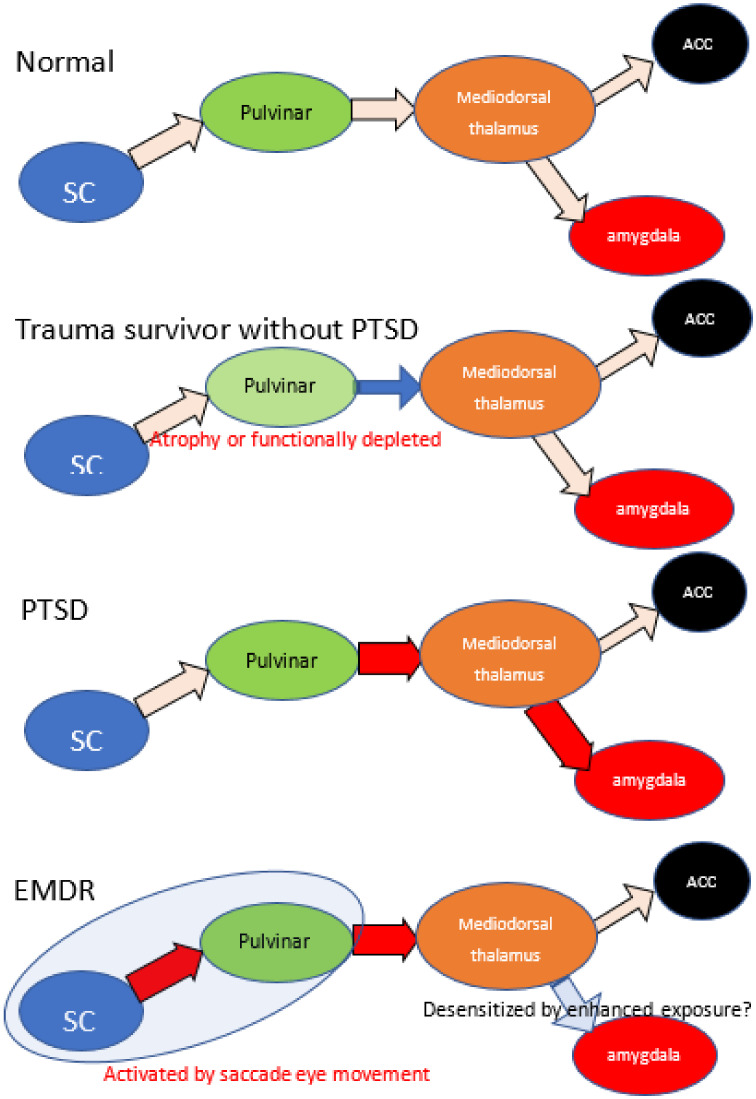
The schematic summary of retinotectal pathway and hypothetic mechanism of EMDR. Because of reports low morbidity rate of trauma survivors with smaller pulvinar, atrophy or dysfunction of pulvinar may have brought prevention of progression in PTSD. PTSD patients indicate enhanced connectivity between thalamus and amygdala, depleted between thalamus and ACC. EMDR intervention enhances the activity of retinotectal pathways and promotes the effects of exposure. Red arrow indicates enhanced connectivity and blue arrow indicate reduced. SC: superior colliculus, ACC: anterior cingulate cortex.

**Table 3 ijms-22-01730-t003:** The summary of literature study for the retinotectal pathway.

The Retinotectal Pathway in Fear-Related Learning in the Thalamus	Reference Number
SC directly modulates defense behavior	[69]
SC is activated in social eye contact situations in PTSD patients	[70,71]
Subliminal threat activates SC and periaqueductal gray	[72]
Enhanced connectivity between SC and dorsal lateral prefrontal cortex in dissociative PTSD	[73]
Pulvinar lesions disrupt implicit fear-related visual processing	[75]
Pulvinar and V1 cortex contribute to fear anticipation	[74]
Contribution of retinotectal pathway to implicit fear processing	[76,77]
Contribution of retinotectal pathway to fear learning	[78]
Smaller pulvinar in traumatized control	[79]
Depletion of right pulvinar seed connectivity to sensory area in dissociative PTSD	[80]

**Table 4 ijms-22-01730-t004:** The summary of literature research for therapy for PTSD related to thalamus.

Discussion	Reference Number
Trauma-exposed control showed depleted connectivity between thalamus and postcentral gyrus	[81]
Enhanced connectivity between the centromedial amygdala and pulvinar, and depletion between the basolateral amygdala and SC	[82]
Amygdala−thalamus connection enhanced during recovery process	[83]
**EMDR**	
Contribution of mediodorsal thalamus to fear processing	[64,84,85,86]
Animal EMDR model provides sustained activation between SC and mediodorsal thalamus with fear reducing effects	[87]
EMDR reduced gray matter volume in the left thalamus	[88]
**Hyperbaric Oxygen Therapy (HBOT)**	
HBOT enhances FA in the thalamic radiation, left thalamus, and insula, with improved PTSD scores	[39]
**Oxytocin Administration**	
Enhanced activity in the left thalamus during tasks both in PTSD and controls	[89]
Oxytocin administration decreased connectivity between left thalamus and amygdala in men with PTSD and traumatized controls, but increased connectivity in woman with PTSD	[89]

## Data Availability

No new data were created or analyzed in this study and data sharing is not applicable to this article.

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
