# Peer review of "The Role of the Thalamus in Post-Traumatic Stress Disorder"

_ijms, 2021, doi:10.3390/ijms22041730_

Round 1

Reviewer 1 Report

The present manuscript is a review of the scientific evidences on the thalamic role in the post-traumatic disorder, including the retino-tectal pathway and therapeutic strategies.

The work is nicely written and covers its objectives. I have some minor concerns that I will like that the author addresses them.

First, it is mentioned in different parts of the manuscript that the first intention of the author was to realize a systematic review but the number of articles were not enough. So, it is surprising that in the limitation section the author indicates that this fact could interfere with the validity of this study as a systematic review. If this is not a systematic review, I would suggest the author to remove any mention to this term all along the manuscript.

In the introduction, the author introduces the terms “prolonged exposure” and “exposure therapy” with no definition. It would be interesting to explain their meanings in a way that non-specialist could understand their significance.

Typographics

Pag 1 line 29 remove “also”

Page 1 line 43 remove “is sensitive to stress and”

Author Response

Thank you for your comment and advice. 

This manuscript was planned as a systematic review and the methodology of the literature study follows the PRISMA rules  (Moher et al 2009). However, some of content in the section of discussion was described independently from the systematic bibliographic research and I feel it may limit the relevance and reproducibility of the discussion.  

notice that both reviewers did not present any concerns about the relevance as a systematic review of this manuscript. For this reason, I hope that this article is possible to be treated as a systematic review.  I revised the section of limitation, page 14 line 14 "the validity as a systematic reviewto the "validity and reproducibility of the discussion”. 

I added introduction for PE in the section of introduction in page1 line 25-30.

I also revised typographics as you mentioned.

Reviewer 2 Report

As the thalamus might be a promising area for stress research, the aim of the present study was to conduct a mini literature review of the contribution of the thalamus to the pathogenesis of PTSD.

Overall, I found this review timely, original, well conducted and scientifically sound. I have only some minor suggestions aimed to improve the high quality of the paper and these are outlined below:

1) In the Introduction, the Authors should briefly explain what is PTSD and what are the risk factors for developing it as IJMS isn't a Journal that is straightly directed to psychiatrists. I suggest to add DSM-V criteria and update some references with appropriate ones (see De Berardis et al. Int J Psychiatry Clin Pract. 2020;24(1):83-87 and Curr Drug Targets. 2015;16(10):1094-106).

2) In Results, page 2, line 24, the Author cited "Figure 1", but, in the paper I don't see any figure. Please correct.

3) I believe that a figure depicting thalamus connections and neurotransmission involved would be greatly appreciated by me and readers in order to make the review' hypothesis improved by a graphical representation. 

4) I believe that also the treatment with prazosin can target the thalamus. Please briefly discuss this point.

Author Response

Thank you for your positive comments. 

1) I added brief sentences to explain PTSD and DSM-V criteria in the frontal part of introduction. I also cited the article (De Berardis et al. Int J Psychiatry Clin Pract. 2020;24(1):83-87for describing TAS-20 and suicide attempt in PTSD patients. I also cited the article in the section of discussion: De Berardis et al. Curr Drug Targets. 2015;16(10):1094-106). 

2),3) In the revised version, figures are included in the manuscript body. Figure 5 is a schematic summery for my hypothesis.

4) Thank you for your advice, I added brief sentences in the discussion, page13 line 33-39.

 Noradrenergic system is intimately connected to the thalamus and it is interesting and promising area for future research seeking the relationship between the noradrenergic modulation and alteration of the thalamus in PTSD patients. 

I also cited  the article: De Berardis et al. Curr Drug Targets. 2015;16(10):1094-106).